



# Geometric accuracy assessment of global coarse resolution satellite data sets: a study based on AVHRR GAC data at the subpixel level

**Xiaodan Wu[1, 2], Kathrin Naegeli[2], and Stefan Wunderle[2]**

[1]College of Earth and Environmental Sciences, Lanzhou University, Lanzhou 730000, China

[2]Institute of Geography and Oeschger Center for Climate Change Research, University of Bern, Hallerstrasse 12, CH-3012 Bern, Switzerland

Correspondence to: Xiaodan Wu (wuxd@lzu.edu.cn)

**Abstract**: AVHRR GAC (Global Area Coverage) data provide daily global coverage of the Earth, which are widely used for global environmental and climate studies. However, their geolocation accuracy has not been comprehensively evaluated due to the difficulty caused by onboard resampling and the resulting coarse resolution, which hampers their usefulness in various applications. In this study, a Correlation-based Patch Matching Method (CPMM) was proposed to characterize and quantify the AVHRR GAC geo-location accuracy at the subpixel level. This method is not limited to landmarks and not suffer from errors caused by false detection due to the effect of mixed pixels, thus enables a more robust and comprehensive geometric assessment. Data of NOAA-17, MetOp-A, and MetOp-B satellites were selected to test the geocoding accuracy. The three satellites predominately present West shifts in the across-track direction, with average values of -1.69 km, -1.9 km, -2.56 km and standard deviations of 1.32 km, 1.1 km, 2.19 km for NOAA-17, MetOp-A, and MetOp-B, respectively. The large shifts and uncertainties are partly induced by the larger satellite zenith angles (SatZ) and partly due to the terrain effect, which is related to SatZ and becomes apparent in the case of large SatZ. It is thus suggested that GAC data with SatZ less than 40° should be preferred in applications. The along-track geolocation accuracy is clearly improved compared to the across-track direction, with average shifts of -0.7 km, -0.02 km, 0.96 km and standard deviations of 1.01 km, 0.79 km, 1.70 km for NOAA-17, MetOp-A, and MetOp-B, respectively. The data can be accessed from http://www.esa-cloud-cci.org/ (Stengel et al., 2017) and https://ladsweb.modaps.eosdis.nasa.gov/ (Didan, 2015).

## 1 Introduction

Advanced Very High Resolution Radiometer (AVHRR) data provide valuable data sources with a near daily global coverage to support a broad range of environmental monitoring researches, including weather forecasting, climate change, ocean dynamics, atmospheric soundings, land cover monitoring, search and rescue, forest fire detection, and many other





applications (Van et al., 2008). The unique advantages of AVHRR sensors is their long history
dating back to the 1980s and thus enabling long-term analyses at climate-relevant time scales
that cannot be covered by other satellites. However, AVHRR data are rarely used at the full
spatial resolution for global monitoring due to the limited data availability (Pouliot et al., 2009;
Fontana et al., 2009). Instead, the Global Area Coverage (GAC) AVHRR dataset with a reduced
spatial resolution is generally employed in long-term studies at a global or regional perspective
(Hori et al., 2017; Delbart et al., 2006; Stöckli et al., 2004; Moulin et al., 1997).

However, there are several known problems with the geo-location of AVHRR GAC data,

which have a profound impact on their application.  (1) The drift of the spacecraft clock results
in errors in the along-track direction (Devasthale et al., 2016). Generally, an uncertainty of 1
second approximately induces an error of 8 km in this direction. (2) Satellite orientation and
position uncertainties influence the projection of the satellite geometry to the ground, which
leads to errors in both along-track and across-track directions. (3) Earth surface elevation
aggravates distortions in the across-track direction (Fontana et al., 2009). Without navigation
corrections, the spatial misplacement of the GAC scene caused by these factors can be up to
25-30 km occasionally (Devasthale et al., 2016).

For geocoding of AVHRR data, a two-step approach is usually used: 1) geocoding based

on orbit model, ephemeris data, and time of onboard clock (Van et al., 2008), achieving an
accuracy within 3-5 km depending on the accuracy of orbit parameters and model (Khlopenkov
et al., 2010); 2) using any kind of ground control points (GCPs) (e.g., road or river intersections,
coastal lines) to improve geocoding (Takagi, 2004; Van et al., 2008). Additionally, in order to
eliminate the ortho-shift caused by elevations, an orthorectification would be needed (Aguilar
et al., 2013; Khlopenkov et al., 2010). The dataset used in this study is from the ESA (European
Space Agency) cloud CCI (Cloud Climate Change Initiative) project, which has corrected clock
drift errors by coregistration of AVHRR GAC data with a reference dataset, and showed
improved navigation by fitting the data to coastal lines.

Unlike the Local Area Coverage (LAC) data with a full spatial resolution of AVHRR, GAC

data are sampled on board the satellite in real-time to generate reduced resolution data (Kidwell,
1998). This is achieved by averaging values from four out of five pixel samples along a scan
line and eliminating two out of three scan lines, resulting in a spatial resolution of 1.1 km × 4
km along the scan line with a 3 km distance between pixels across the scan line. Therefore, the
nominal size of a GAC pixel is 3 km × 4.4 km. It is important to note that the spatial resolution
of GAC data also depends on the satellite zenith angle (SatZ). Because of the large swath width,
the spatial resolution of LAC decreases to 2.4km by 6.9 km at the edge of the swath (D'Souza
et al., 1994). With the selection process for GAC, the GAC resolution is also much worse than
4 km. Furthermore, the onboard resampling process of GAC data makes the orthorectification
not feasible, which results in lowering of geolocation accuracy in the across-track direction.
The final quality of AVHRR GAC data has not been quantified and we, therefore, make an
attempt to assess their geolocation accuracy, particularly over terrain areas.



There are generally three approaches to assess the non-systematic geometric errors of satellite images: (1) the coastline crossing method (CCM) which detects the coastline in the along-track and across-track directions through a cubic polynomial fitting (Hoffman et al., 1987); (2) the land-sea fraction method (LFM) which develops a linear radiance model as a function of land-sea fraction, land and sea radiance, and then finds the minimum difference between model-simulated and instrument-observed radiance by shifting the pixels in along-track and across-track directions; (3) the coregistration method which computes the difference or similarity relative to a reference image (Khlopenkov et al., 2010). The abilities of these methods in characterizing the geometric errors are limited to certain conditions. The CCM is subject to the structure of coastline. Although the LFM works better on complex coastlines but depends on the accuracy of the land-sea model. The coregistration method is usually applied to high-resolution visible and infrared images (Wang et al., 2013; Wolfe et al., 2013). When it comes to coarse resolution data with several kilometers, the main difficulty arises from false detection due to the effect of mixed pixels. The geometric accuracy is important as even small geometric errors can lead to significant noises on the retrieval of surface parameters, such as NDVI, LAI, and albedo, which mask the reality or bias the final results and conclusions (Khlopenkov et al., 2010; Arnold et al., 2010). For instance, anomalous NDVI dynamics during the regeneration phase of forest fire-burnt areas can be explained by the imprecise geolocation of the data set used (Alcaraz-Segura et al., 2010). Therefore, it is critical to develop a rigorous geometric accuracy assessment method in order to ensure the effectiveness of AVHRR GAC data in the generation of climate data records (CDR) (Khlopenkov et al., 2010; Van et al., 2008).

Based on the idea of the coregistration method, this study proposes a method named Correlation-based Patch Matching Method (CPMM), which is capable of quantifying the geometric accuracy of coarse resolution satellite data available as fundamental climate data records (FCDR) for global applications (Hollmann et al., 2013). We show the procedure based on AVHRR GAC data, which are compiled for the ESA CCI cloud project (Stengel et al., 2017) and are now also used for the ESA CCI+ snow project. The assessment is conducted at the sub-pixel level and not affected by the mixed pixel problem. This method is applied to some test data from NOAA-17, MetOp-A, and MetOp-B, respectively. Furthermore, the potential factors that cause geometric distortions are explored and discussed. Although the band-to-band registration (BBR) accuracy assessment is an important aspect for such multi-spectral images, it is not a focus of this study, since the BBR accuracy of AVHRR has been comprehensively evaluated by a previous study (Aksakal et al., 2015).

## 2 Data and geographical regions of interest

### 2.1 Satellite data

AVHRR is a multipurpose imaging instrument aboard on the NOAA satellite series since 1978 and the Meteorological Operational Satellites (MetOp) operated by EUMETSAT since



2006, delivering daily information of the Earth in the visible, near-infrared, and thermal
wavelengths. They provide observations from 4 to 6 spectral bands, depending on the
generation of AVHRR sensors. This study only focuses on the AVHRR GAC data observed by
NOAA-17 (AVHRR-3 generation), MetOp-A, and MetOp-B. The spectral characteristics of the
AVHRR sensors on board these three platforms are the same and summarized in Table 1. Since
the spatial resolution of AVHRR GAC data is often considered to be 4 km (Fontana et al., 2009),
the analysis in this study was conducted at the 4 km level using the data acquired on August 13,
2003 for NOAA-17 and March 12, 2017 for MetOp-A and MetOp-B.
**Table 1**. Spectral characteristics of AVHRR sensors

| Band | Wavelength (µm) | Application |
|---|---|---|
| 1 | 0.58–0.68 (VIS) | Cloud mapping, vegetation and surface characterization |
| 2 | 0.72–1.00 (NIR) | Vegetation mapping, water body detection |
| 3a* | 1.58–1.64 (MIR) | Snow and Ice classification |
| 3b* | 3.55–3.93 (MIR) | Cloud detection, Sea/Land surface temperature, |
| 4 | 10.30–11.30 (TIR) | Cloud detection, Sea/Land surface temperature, |
| 5 | 11.50–12.50 (TIR) | Cloud detection, Sea/Land surface temperature |

*Note: Channel 3a is only used continuously on NOAA-17 and MetOp-A. On-board MetOp-B channel 3a was only
active during a limited time span.
From a standpoint of geometric accuracy assessment, the reflectances in band 1 and 2 were
employed in this study. However, these two bands are not only affected by the atmosphere but
also by the earth surface anisotropy characterized by the bidirectional reflectance distribution
function (BRDF) (Cihlar et al., 2004). Given the fact that BRDF effects can be reduced through
the calculation of vegetation indices such as NDVI (Lee & Kaufman, 1986), the NDVI is
employed in this study, which is derived from the reflectance in band 1 and 2 according to
Equation (1).

$$NDVI = \frac{R_2 - R_1}{R_2 + R_1} \tag{1}$$

where $R_1$ and $R_2$ refer to the reflectance in band 1 and 2, respectively. It is important to note
that during the process of generating NDVI, the atmospheric and BRDF corrections were not
performed. But it is expected that such effects originating from these omissions are of minor
influence, because the method of this study is based on correlation analysis and does not rely
on absolute values of NDVI. Another advantage of using NDVI is that it has higher contrast
between different land cover types, such as vegetation/no-vegetation, snow/no-snow, etc.
Furthermore, in order to investigate the effect of off-nadir viewing angle on geometric accuracy,
the SatZ data of AVHRR were also extracted.
Ideally, the referenced data in geometric quality assessment should meet the required
accuracy of 1/3 field of view (FOV) (WMO and UNEP, 2006), and also satisfy the accuracy



requirement of an order of magnitude better than one-tenth of the image spatial resolution
(Aksakal, 2013), which means 400 m for the AVHRR GAC data. The NDVI provided by
MOD13A1 V006 product was introduced as a source of reference data to perform the geometric
quality assessment, because the sub-pixel accuracy of MODIS product is sufficient to satisfy
this requirement (Wolfe et al., 2002). The high geolocation accuracy of MODIS products was
achieved by using the most advanced data processing system, which has updated the models of
spacecraft and instrument orientation several times since launch. Consequently, the various
geolocation biases resulted from instrument effects and sensor orientation are removed (Wolfe
et al., 2002). The NDVI data with the date corresponding to that of AVHRR GAC data, were
obtained from the Level-1 and Atmosphere Archive & Distribution System (LAADS)
Distributed Active Archive Center (DAAC)  (https://ladsweb.modaps.eosdis.nasa.gov/) with
the sinusoidal projection at a spatial resolution of 500 m and a temporal resolution of 16-day.
The detailed description of the MOD13A1 V006 product can be found in Didan (2015).
**2.2 Geographical regions of interest**

The purpose of this study is not only to assess the geolocation accuracy of 4 km AVHRR
GAC data, but also to explore the potential impact factors related to geolocation accuracy.
Therefore, the investigations were made at different latitudes and longitudes, at different
locations with different SatZ, for different land covers, as well as different topographies. The
swaths covering parts of Europe (including the alpine mountain) and Africa were used since
they fit the study needs (Fig. 1). Investigations were based on six regions of interest (ROI) as
shown in Figs. 1 and 2. The ROIs from 1 to 6 enable us to investigate the geolocation accuracy
at different SatZ, topography, as well as latitudes and longitudes. Their locations and extents
are consistent for the scenes from NOAA-17 and MetOp-A (Fig. 1), which enables the
comparison of geolocation accuracy between these two sensors. The size of ROI was attempted
to be set as large as possible in order to get more significant and comprehensive results. On the
other hand, areas covered by cloud and water have to be avoided, resulting in the different sizes
of these ROIs. Half of the ROIs (ROIs 2, 4, 6) serve as a good example for a typical
mountainous areas on Earth. The other half of ROIs (ROIs 1, 3, 5), on the other hand, mainly
cover relatively flat areas. Since the NOAA-17 scene was almost unaffected by cloud, another
ROI (ROI 7) was selected to check the geolocation accuracy at nadir. The MetOp-B scene was
influenced by cloud but served as a good example to illustrate the combined effect of
topography and large SatZ (Fig. 2). Although there are also 6 ROIs selected, their sizes and
extents are totally different from the above two scenes.  In order to include the terrain area, two
subsets were used (Figs. 2a and c). Each grid in the ROI represents the minimum unit (namely
the patch) based on which we conduct the geometric quality analysis.

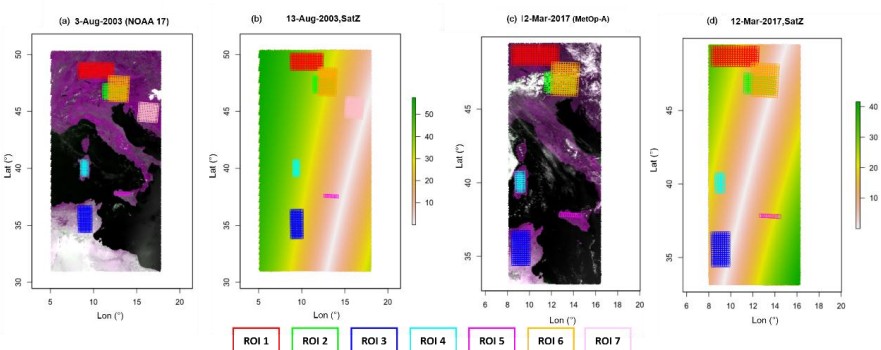

**Figure 1.** The study area and the distribution of ROIs. (a) and (c) are the composite maps of bands 2-1-1 of AVHRR GAC data on August 13, 2003 and March 12, 2017, respectively. (b) and (d) are their corresponding SatZ, respectively.

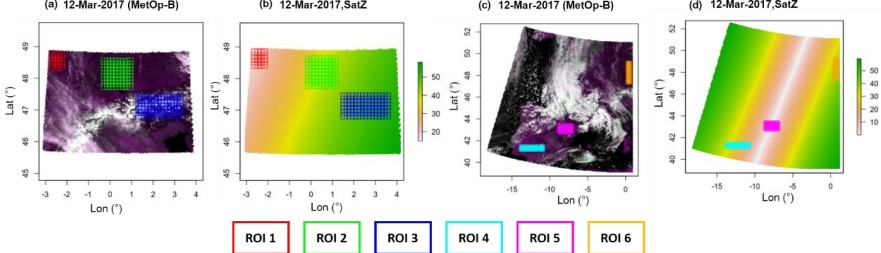

**Figure 2.** The study area and the distribution of ROIs on March 12, 2017. (a) and (c) are the composite maps of bands 2-1-1 subset 1 and 2, respectively. (b) and (d) are their corresponding SatZ, respectively.

# 3 Methodology

The assessment was performed by comparing the AVHRR GAC scenes with geo-located reference data, i.e. MOD13A1 (V006). An approach named Correlation-based Patch Matching Method (CPMM) is proposed to find the best match between small image patches taken from the reference images and the AVHRR GAC images. This method is expected to be more suitable for the geometric accuracy assessment of coarse resolution images than the current methods, i.e. the CGM, LFM, and co-registration using shorelines. Because it is not limited to a certain landmark such as a lake or sea shoreline, and thus enables a more comprehensive assessment over different areas in the satellite scene. Moreover, this method does not suffer from errors caused by false detection due to the effect of mixed pixels because it is applied directly on the pixel values. The framework of CPMM is shown in Fig. 3, and the detailed description of this method is provided below.



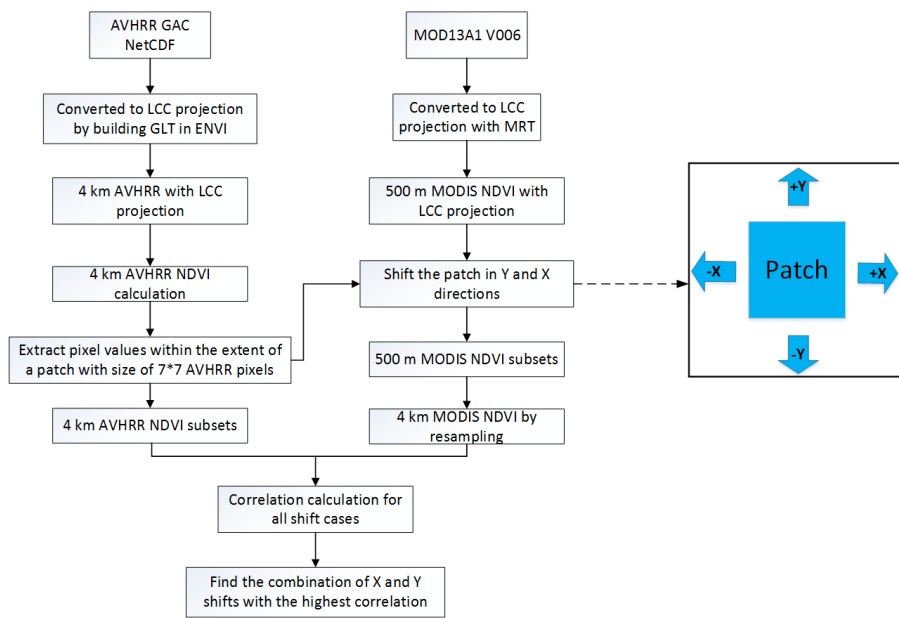

**Figure 3.** Flowchart of the Correlation-based Patch Matching Method (CPMM).

### 3.1 Satellite data processing

196 The AVHRR GAC data set is stored in a Network Common Data Format (NetCDF), with

197 latitude and longitude assigned to each pixel. In order to achieve a higher accuracy of image

198 matching, the data need to be reprojected. The AVHRR GAC scene was reprojected into the

199 Lambert Conformal Conic (LCC) projection by building the Geographic Lookup Table (GLT)

200 using the latitude and longitude data in ENVI. The spatial resolution of the AVHRR GAC map

201 in the LCC projection is 4 km. Based on the reprojected data, the NDVI was calculated using

202 the band combinations as indicated by Eq. (1). Similarly, the NDVI band of MOD13A1 in the

203 HDF format was extracted and converted to LCC projection from its raw sinusoidal projection

204 using the MODIS Reprojection Tool (MRT). The nearest neighbor (NN) resampling scheme

205 was employed in this procedure. The spatial resolution of the MODIS NDVI in the LCC

206 projection is 500 m. Thus, the geometric assessment is performed at the 4 km resolution of

207 AVHRR NDVI based on the 500 m MODIS NDVI data.

### 3.2 Patch matching and geometric assessment

209 In the process of matching the AVHRR GAC data with reference MODIS data, a patch

210 size of 7 × 7 AVHRR pixels (corresponding to approximately 28 km × 28 km) was used. These

211 patches were distributed in each ROI as shown in Figs. 1 and 2, with an interval of 4 pixels in

212 the along-track (Y-) and across-track (X-) direction. The sizes of the patch and interval were

213 determined based on the following aspects: the size of the patch should contain enough pixels

214 to support a robust correlation estimation, but at the same time, should not be too large in order



to investigate the potential influencing factors related to the geometric accuracy, and get enough
results from these patches to attain a more significant and comprehensive conclusion. Similarly,
the size of the interval should enable the disparity between different patches on one hand and
on the other hand a large number of patches within the extent of each ROI. The chosen size has
proven to be most ideal for these criteria during the test of different patch size.

For each patch in the ROI, the AVHRR GAC data within the patch were extracted. Then

the patch was shifted in the Y- and X-direction as indicated by the blue arrows in Fig. 3. Shifts
were conducted stepwise in order to achieve sub-pixel accuracy, beginning with only 500 m
and adding up to 8 km (i.e., ± 2 pixels) at a step of 500 m (equivalent to the MODIS pixel size)
in any direction of Y- and X-combination. Consequently, 33×33 combinations of X- and Y-
shifts have been simulated. For each shift, the MODIS NDVI pixels within the extent of the
patch were extracted and aggregated to 4 km by spatial averaging.  Afterwards, the correlation
between the 4 km rescaled MODIS NDVI and the 4 km AVHRR NDVI was calculated for each
shift in X- and Y-direction. The displacement of one patch was indicated by the shift
combination with the best correlation, which means the geolocation accuracy of the patch. In
this way, the geolocation errors were transformed into the across-track and along-track
directions at the sub-pixel level for correlation with possible error sources.

It is expected that the results from each patch are different. Therefore, the general accuracy

of each ROI was determined by summarizing the measured shifts of each respective patch
statistically. Here, the histogram was employed to show the distribution of geometric errors in
the across-track and along-track directions. And the quantitative indexes, such as the number
of patches, their mean and standard errors, were calculated.  The averaging is expected to reduce
the uncertainties caused by random factors and produce accurate shift measurement estimates
(Bicheron et al., 2011). The final shifts of the scene were calculated by averaging the measured
shifts of all patches on the scene.

### 3.3 Influence factor

The influence of potential variables on the geometric accuracy was studied, including

SatZ, topography, latitudes, and longitude. To achieve this, the information of these factors were
also extracted for each patch on the scene. The geometric errors induced by SatZ were
highlighted by checking the relationship between errors and SatZ.  The effect of topography
was investigated by checking the relationship of geometric errors in the across-track direction
over terrain areas compared to relatively flat areas. The effect of latitudes and longitude was
determined by analyzing their relationship with measured shifts on the along-track and across-
track directions, respectively.

## 4   Results and discussions

Fig. 4 shows the correlation distribution over the 33 × 33 shifted cases within ± 8 km range

at a step change of 500 m. Here, only one patch is extracted from each respective scene to
illustrate the results. Each grid in Fig. 4 represents a shift combination case, which is indicated
by the location of the grid away from the center. Then the geolocation errors can be transferred
into distances in kilometer (km) by multiplying the location of a grid with 500 m. The center
of each subfigure depicts the case in which the location of the patch on the reference scene is
exactly overlapped with that on the AVHRR scene. The results are visualized for one example
showing the spatial distribution of correlation between the MODIS reference scene and the
AVHRR data (Fig. 4). The color coding indicates a high correlation in dark green and reddish-
white colors indicate low correlation values. An almost perfect match is shown in Fig. 4b, where
the dark green area is nearly centered at the coordinates (0, 0). From Fig. 4a, it can be found
that the patch on the NOAA-17 scene shows geolocation errors of -1 km and 0 km in the along-
track and across-track directions, respectively. The Fig. 4b indicates a geolocation error of 0
km and -0.5 km in the along-track and across-track directions respectively for the patch on the
MetOp-A scene. And Fig. 4c indicates that the patch on the MetOp-B scene shows a geometric
error of 2 km in the along-track direction and -5.5 km in the across-track direction. However,
these figures show only the results of one single patch. The final results are based on a large
number of samples to be statistically significant.

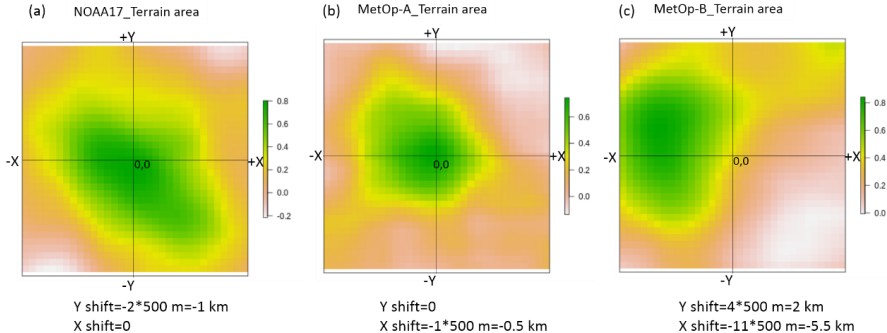


**Figure 4.** Variations of the correlation with respect to each shift combination. Only the results of one
patch from the NOAA-17 (left), MetOp-A (middle), and MetOp-B (right) scenes are shown for
conciseness.
**4.1 Geocoding accuracy**
The geolocation shifts of each patch are slightly different as shown in Figs. 5-7. The +y
indicates a shift to the North and +x indicates a shift to the East (minus sign indicates opposite
directions). The statistical indicators such as the mean value of shift (Mean), the standard
deviation of shift (StdDev) and the number of patches (N), are derived from the estimated shift
values of all patches within the extent of the corresponding ROI.
As shown in Fig. 5, it can be seen that the scene of NOAA-17 generally shows West shifts
in the across-track direction, since the majority of patches in all ROIs show negative shifts.
Nevertheless, the magnitudes of shifts for different ROIs vary from one to another. ROI 2 shows
the smallest shift with a mean value of -0.76 km, with most shifts concentrated around -1 (Fig.

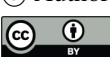

5b). The ROIs 6 and 5 indicate the second smallest shifts, with still weak magnitudes of -1.33
and -1.35, respectively. Most of their shifts are distributed between -2 and 0 (Figs. 5f and e).
The ROIs 7, 3, 1, 4 show slightly larger mean shifts but are still with the magnitudes of less
than 2.5 km. These results are unexpected, because the ROIs (ROIs 2 and 6) over terrain areas
are with smaller shifts than those (ROIs 7, 3, 1, 4) over relatively flat areas in the across-track
direction. One possible reason is that the SatZ for ROIs 2 and 6 are not large (less than 40°)
(Fig. 1b) so that the terrain effect on geolocation accuracy is counterbalanced by the small SatZ.
This also indicates that the influence of small SatZ may be stronger than the terrain effect. But
it is surprising that the ROI 7 (Fig. 5g), which is located at the nadir area (Fig. 1b), shows even
larger shifts than other ROIs (ROIs 2, 6 and 5) with relatively larger SatZ. On the other hand,
ROI 7 shows the most stable behavior, indicated by the smallest StdDev of 0.77. Other ROIs
present relatively large, but still acceptable variations with StdDev ranging from 0.97 to 1.41
(Figs. 5a-g).

When combining the results of all ROIs together (Fig. 5h), the shifts in the across-track

direction generally follow an approximately normal distribution with a mean value of -1.69 and
a standard deviation of 1.32. Nearly 91% of the shifts are within the range of $\pm 3$ km, and the
great majority (97%) of the shifts lay within a range of $\pm 4$ km. The number of patches (N=759)
is assumed to be sufficient to ensure reliability and robustness of the results and the reduction
of the influence of random factors.

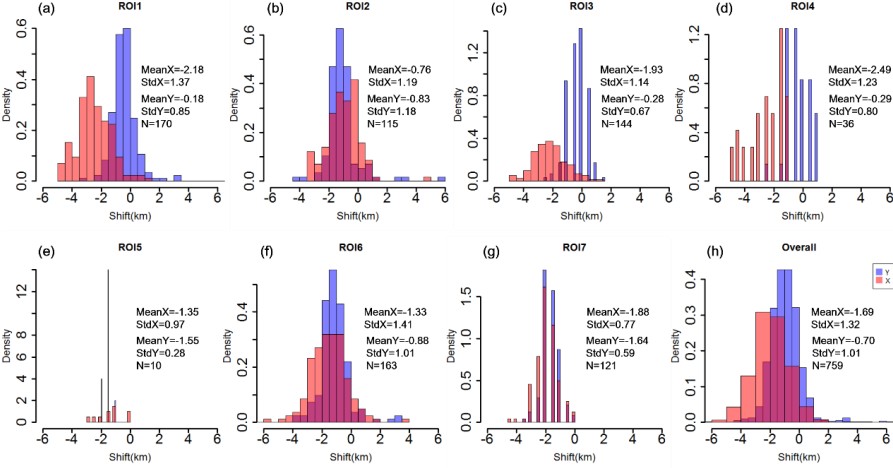


**Figure 5.** The distribution of shifts in the across-track (X) and along-track (Y) directions over different
regions for NOAA-17 scene. The unit of the shift is km.

The shifts in the along-track direction are mainly negative throughout these ROIs,

indicating that the NOAA-17 scene is dominated by South shifts in the along-track direction.
Nevertheless, a considerable number of patches also show slight North shifts over ROIs 1, 3
and 4 (Figs. 5a, c and d), where the shifts are distributed around 0 with mean values of -0.18, -



0.28 and -0.29, respectively. These shifts are generally small in these three regions given that
the maximum shift is no more than 3.5 km (Table 2). In contrast, the ROIs 2, 5, 6 and 7 present
systematic shifts to the South, which are mostly distributed within the range of -2 to 0 km, with
mean values of -0.83, -1.55, -0.88 and -1.64, respectively (Figs. 5b, e, f and g). The large
differences in the distribution of shifts over different ROIs demonstrate that the shifts in the
along-track direction are dependent on the region. It is interesting to find that ROI 7 still shows
the smallest StdDev of 0.59 when excluding ROI 5 due to its very small number of patches.
This indicates that ROI 7 also shows the smallest uncertainty in the along-track direction. And
this may be associated with its smallest SatZ among all investigated ROIs. When combining
the results of different ROIs (Fig. 5h), the overall shifts in the along-track direction
approximately obey a normal distribution, with an average of -0.70 and a standard deviation of
1.01. Nearly 70% of them are within the range of $\pm 1$ km, and only a small part (1.5%) show
values larger than 3 km.

Furthermore, it can be stated that the distribution of shifts in the along-track direction is

less widely spread than that in the across-track direction, demonstrating the smaller uncertainty
of geocoding in the along-track direction, as indicated by the smaller StdDev values throughout
these ROIs (Table 2). Moreover, the geolocation errors in the across-track direction are greater
than the along-track direction (Fig. 5), which is expected due to the applied clock drift
correction.

**Table 2**. Summary of the results for the scene of NOAA-17. The unit of the shift is km.

| ROI | Min(X) | Max(X) | Mean(X) | StdDev(X) | Min(Y) | Max(Y) | Mean(Y) | StdDev(Y) | N |
|---|---|---|---|---|---|---|---|---|---|
| 1 | -5 | 7 | -2.18 | 1.37 | -3.5 | 3.5 | -0.18 | 0.85 | 170 |
| 2 | -3.5 | 5 | -0.76 | 1.19 | -4.5 | 6 | -0.83 | 1.18 | 115 |
| 3 | -5 | 1.5 | -1.93 | 1.14 | -2.5 | 1.5 | -0.28 | 0.67 | 144 |
| 4 | -5 | -1 | -2.49 | 1.23 | -2.5 | 1 | -0.29 | 0.80 | 36 |
| 5 | -3 | 0 | -1.35 | 0.97 | -2 | -1 | -1.55 | 0.28 | 10 |
| 6 | -7.5 | 4 | -1.33 | 1.41 | -4 | 3.5 | -0.88 | 1.01 | 163 |
| 7 | -4.5 | 0 | -1.88 | 0.77 | -3.5 | 0 | -1.64 | 0.59 | 121 |
| Overall | -7.5 | 7 | -1.69 | 1.32 | -4.5 | 6 | -0.70 | 1.01 | 759 |

Similar to the results of NOAA-17, MetOp-A scene mainly present West shifts in the

across-track direction, indicated by the widely distributed negative values throughout these
ROIs (Figs. 6a-f). These shifts are basically concentrated around -2, however, the ROIs 2 and
6 located in the terrain areas, show smaller average shifts (-1.68 and -1.82, respectively) than
those of ROIs 1 and 3 (-2.25 and -1.94, respectively) over the relatively flat areas. This is
understandable since the ROIs 2 and 6 are closer to the nadir area (Fig. 1d). And this align with
the results from NOAA-17, where the influence of SatZ is also stronger than the terrain effect.
Although the ROIs 5 and 4 show the smallest average shifts (-0.72 and -1.45, respectively) in
the across-track direction, their results may be biased due to the smaller number of analyzed
patches. It is interesting to find that ROI 3, which is almost located in the nadir area, still shows
the least uncertainty, indicated by the smallest StdDev of 0.67. Furthermore, all ROIs close to
the nadir area are characterized by small StdDevs (0.8 and 1.03 for ROIs 2 and 6, respectively)
compared to ROIs located further away from the nadir area (1.29, 2.05, 1.37 for ROIs 1, 4, 5,
respectively).  These results demonstrate that SatZ plays a crucial role in determining the
uncertainty of the shifts in the across-track direction. This conclusion also agrees with previous
research conducted by Aguilar et al. (2013). When combining the results of all ROIs (Fig. 6g),
the shifts approximately follow a normal distribution, with an average of -1.90 and a standard
deviation of 1.1. Most of the patches (94%) are within the range of ±3 km, and nearly 98% of
them are with shifts less than ±4 km.
Since ROIs 1-6 on the MetOp-A scene are identical to those on NOAA-17 scene in terms
of spatial extents, their shifts in the across-track direction are generally comparable. When
excluding the results of ROIs 4 and 5, the ROIs on the MetOp-A scene generally show larger
average shifts but smaller StdDevs than the NOAA-17 scene in the across-track direction (see
Table 2 and 3).  However, it does not necessarily mean that the MetOp-A scene has a smaller
uncertainty than NOAA-17 scene in the across-track direction, because the ROIs on the MetOp-
A scene are slightly closer to the nadir area than those on the NOAA-17 scene (Figs. 1b and d).
Given the larger SatZ and the smaller average shifts of NOAA-17 scene, it is reasonable to
conclude that the NOAA-17 scene shows a slightly better geolocation accuracy than the
MetOp-A scene in the across-track direction.

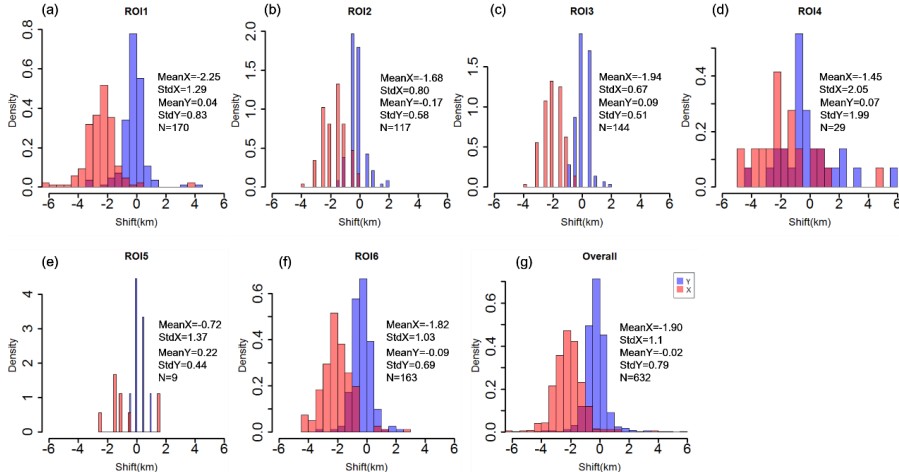


**Figure 6.** The distribution of shifts in the across-track (X) and along-track (Y) directions over different
regions for MetOp-A scene. The unit of the shift is km.

Looking at the shifts in the along-track direction, the MetOp-A scene does not show strong
systematic North or South shifts,  but rather a general distribution of the shifts around 0 (Figs.
6a-f). The shifts are generally small within a range of $\pm 1$ km, with StdDevs less than 0.83
except for ROI 4. Furthermore, ROIs 2, 3 and 6 that are located close to the nadir area exhibit
smaller StdDevs than those located further away from the nadir area when excluding ROI 5 due
to its very small number of patches. This further indicates that SatZ also determines the
uncertainty of shifts in the along-track direction. When combining the results of all ROIs (Fig.
6g), the shifts also display a nearly normal distribution, with an average of -0.02 and a StdDev
of 0.79. Nearly 94% of the shifts are within the range of ±1 km and almost all of them (98%)
are distributed within the range of ±2 km. It can be found that the shifts in the along-track
direction are obviously smaller and more centralized than those in the across-track direction.
This can be further confirmed by the consistently smaller StdDev values in the along-track
direction than those in the across-track direction as shown in Table 3.
**Table 3**. Summary of the results for the scene of MetOp-A. The unit of the shift is km.

| ROI | Min(X) | Max(X) | Mean(X) | StdDev(X) | Min(Y) | Max(Y) | Mean(Y) | StdDev(Y) | N |
|---|---|---|---|---|---|---|---|---|---|
| 1 | -7 | 4 | -2.25 | 1.29 | -3.5 | 4.5 | 0.04 | 0.83 | 170 |
| 2 | -4 | 0 | -1.68 | 0.80 | -1.5 | 2 | -0.17 | 0.58 | 117 |
| 3 | -4 | -0.5 | -1.94 | 0.67 | -1 | 2 | 0.09 | 0.51 | 144 |
| 4 | -5 | 5 | -1.45 | 2.05 | -4.5 | 6 | 0.07 | 1.99 | 29 |
| 5 | -2.5 | 1.5 | -0.72 | 1.37 | -0.5 | 1 | 0.22 | 0.44 | 9 |
| 6 | -4.5 | 3 | -1.82 | 1.03 | -3.5 | 2.5 | -0.09 | 0.69 | 163 |
| Overall | -7 | 5 | -1.90 | 1.10 | -4.5 | 6 | -0.02 | 0.79 | 632 |

By comparing Figs. 6a-f with Figs. 5a-f, it becomes obvious that large differences exist

between the shifts in the along-track direction of MetOp-A and NOAA-17 scenes. In the first
place, systematic South shifts occur on the NOAA-17 scene but not on the MetOp-A scene.
Secondly, the magnitudes of shifts on the MetOp-A scene are generally smaller than those on
the NOAA-17 scene, as the former are concentrated around 0 while the latter are concentrated
around -1. Thirdly, the distribution of shifts is more centralized for the MetOp-A scene
compared to the NOAA-17 scene, except for ROIs 4 and 5. This can further be proved by the
smaller StdDev values for MetOp-A (Table 3) than those for NOAA-17 (Table 2). Therefore, it
can be concluded that the MetOp-A scene shows a better geolocation accuracy and less
uncertainty than the NOAA-17 scene in the along-track direction.

Similar to the scenes of NOAA-17 and MetOp-A, the MetOp-B scene generally shows

West shifts in the across-track direction, indicated by the predominant occurrence of negative
values (Figs. 7a-f). Nevertheless, unlike the results for the terrain areas on NOAA-17 and
MetOp-A scenes, the ROI 3 located in the terrain area on the MetOp-B scene (Fig. 2a), shows
the largest shifts throughout these ROIs with an average of -4.69 in the across-track direction.
Furthermore, the magnitudes of these shifts are characterized by even larger values than 6 km
(Fig. 7c). This is most probably caused by the combined effect of topography and large SatZ
(Fig. 2b). Significant terrain effects appear only in the case of SatZ larger than 40° as shown in
Fig. 2b. This finding agrees with the previous study by Fontana et al. (2009), who demonstrated
that the errors in across-track direction result from the intertwined effects of observation
geometry and terrain elevation. Nevertheless, ROI 5 that is located in the nadir area (Fig. 2d),
shows the smallest average shift of -1.29 but the largest standard deviation of 2.51 (Fig. 7e).
The largest StdDev is attributed to the fact that a considerable number of shifts exhibit values
of ±6 km. As shown in Fig. 2c, the main reason for these large and unstable shifts may be the
presence of thin clouds or cloud shadows in this region. By comparing the results of ROIs 4
and 5 with smaller SatZ against ROIs 2, 3, 6 with larger SatZ (Figs. 2b and d), it can be stated
that the shifts with smaller SatZ are generally weaker than those with larger SatZ (Figs. 7b-f).
When combining the results of all ROIs (Fig. 7g), the MetOp-B scene shows an average shift
of -2.56 km with a standard deviation of 2.19 in the across-track direction. Only 63% of the
shifts are distributed within the range of ±3 km, and the percentage raises up to 92% within
the range of ±5.5 km.

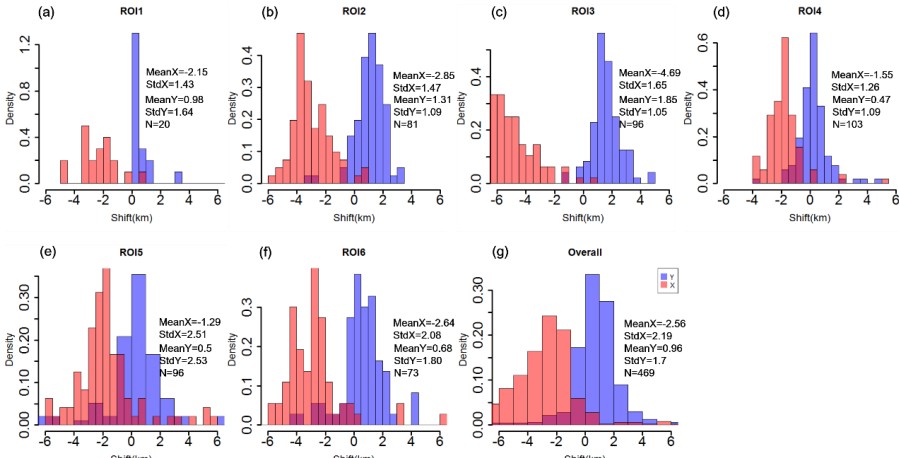


**Figure 7.** The distribution of shifts in the across-track (X) amd along-track (Y) directions over different
regions for MetOp-B scene. The unit of the shift is km.
Since the extent of the ROIs in the MetOp-B scene are not consistent with those on NOAA-
17 and MetOp-A scenes, only their overall performances in the across-track direction are
compared here. By comparing Fig. 7g with Fig. 6g and Fig. 5h, it is obvious that the MetOp-B
scene shows larger shifts and greater uncertainties than NOAA-17 and MetOp-A scenes in the
across-track direction. This is partly due to the larger range of SatZ of these ROIs and partly
due to the worse geolocation accuracy of the MetOp-B scene in the across-track direction.
The MetOp-B scene is dominated by North shifts in the along-track direction, indicated
by the predominantly positive shift values (Figs. 7a-f). It is interesting to find that ROI 3, which
is located at terrain area and with large SatZ, shows the largest shifts with an average of 1.85
km in the along-track direction. Given that terrain does not affect the geolocation accuracy in
the along-track direction, the main cause of the largest shift may be the largest SatZ of ROI 3
among these ROIs. Furthermore, by comparing the results of ROI 4 and 5 with those of ROI
2, 3, 6, it can be found the shifts of ROIs with smaller SatZ are more concentrated around 0
(Figs. 7d and e), while the shifts of ROIs with larger SatZ are more widely spread (Figs. 7b, c,
and f). This manifests that the effect of large SatZ on shifts in the along-track direction cannot





be neglected. When combining the results of all ROIs, the MetOp-B scene shows shifts with an average of 0.96 and a standard deviation of 1.7. Only 52% of the shifts are distributed within the range of $\pm 1$ km, and the percentage raises up to 92% for the range of $\pm 3$ km.

It can be seen that the shifts in the along-track direction are still significantly smaller than those in the across-track direction. Furthermore, the uncertainties of the shifts in the along-track direction are generally smaller than those in the across-track direction, when excluding the results of ROI 1 due to its limited number of patches (Table 4). This further verifies that after removing clock drift errors, the geolocation errors in the along-track direction are generally more accurate and with less uncertainties than the across-track direction.

**Table 4**. Summary of the results for the scene of MetOp-B. The unit of the shift is km.

| ROI | Min(X) | Max(X) | Mean(X) | StdDev(X) | Min(Y) | Max(Y) | Mean(Y) | StdDev(Y) | N |
|---|---|---|---|---|---|---|---|---|---|
| 1 | -5 | 1 | -2.15 | 1.43 | 0 | 7 | 0.98 | 1.64 | 20 |
| 2 | -7.5 | 1 | -2.85 | 1.47 | -3.5 | 3.5 | 1.31 | 1.09 | 81 |
| 3 | -7.5 | 1 | -4.69 | 1.65 | -1.5 | 5 | 1.85 | 1.05 | 96 |
| 4 | -4 | 5.5 | -1.55 | 1.26 | -4 | 5 | 0.47 | 1.09 | 103 |
| 5 | -6 | 7.5 | -1.29 | 2.51 | -7.5 | 7.5 | 0.50 | 2.53 | 96 |
| 6 | -7.5 | 6.5 | -2.64 | 2.08 | -7 | 4.5 | 0.68 | 1.80 | 73 |
| Overall | -7.5 | 7.5 | -2.56 | 2.19 | -7.5 | 7.5 | 0.96 | 1.70 | 469 |

The comparison of Fig. 7g with Fig. 6g and Fig. 5h reveals that the MetOp-B scene is significantly inferior to the MetOp-A scene in terms of the geolocation accuracy in the along-track direction, with the former being concentrated around 1 and the latter around 0. Furthermore, the uncertainty of the shifts of the MetOp-B scene (StdDev=1.7) is much larger than that of the MetOp-A scene (StdDev=0.79). As for the performance of the MetOp-B scene relative to the NOAA-17 scene, it can be found that they are comparable with regard to the magnitude as well as the distribution of the shifts in the along-track direction. However, the MetOp-B scene shows larger uncertainties than NOAA-17.

From the results above, it can be concluded that NOAA-17 and MetOp-A scenes show distinct advantages over the MetOp-B scene in both directions. However, the NOAA-17 scene is slightly better than the MetOp-A scene in the across-track direction, with average shifts of -1.69 for NOAA-17 and -1.90 for MetOp-A, which are both greatly lower than for MetOp-B (-2.56). But the MetOp-A scene shows a distinct advantage over NOAA-17 in the along-track direction, with an average shift of -0.02 for MetOp-A and -0.7 for NOAA-17, which are both lower than for MetOp-B (0.96). In addition to the magnitudes of their shifts, the MetOp-B scene also shows larger uncertainties than NOAA-17 and MetOp-A scenes in both directions.

**4.2 The potential influence factors**

From the above results, it is known that SatZ plays an important role in determining the geolocation accuracy of the satellite scene. To investigate how and to what extent it influences the geolocation accuracy, Fig. 8 displays the shifts in both directions as a function of SatZ for all three satellites. Furthermore, the influences of latitude and longitude on geolocation



accuracy are also explored.

As shown in Figs. 8a-c, it can be seen that the shifts in the across-track direction vary

considerably for all SatZ, and this is particularly evident in the results of MetOp-B (Fig. 8c).
This demonstrates that besides the SatZ effects, the geolocation accuracy is also influenced by
other factors. Furthermore, the spread at each fixed SatZ tends to become larger at larger SatZ
(larger than 20°) (Figs. 8a-b). The large variability of MetOp-B scene shifts at small SatZ (less
than 20°) (Fig. 8c) is mainly due to the effect of thin cloud or cloud shadow as explained before.
Despite the dispersion of the shifts for all SatZ, it can still be found that the shifts in the across-
track direction do not change much when the SatZ is less than 20° (Figs. 8a-b and Table 5). A
slightly decreasing trend (increasing trend of the magnitude) can be observed from 20° to 40°
(Table 5), and becomes more apparent at SatZ larger than 40° (Fig. 8c and Table 5).
Furthermore, it can be found that for small SatZ (less than 20°) the shifts in the across-track
direction are generally concentrated around 2 km for NOAA-17 and MetOp-A scenes (Figs. 8a-
b). With increasing SatZ, the largest magnitudes of shifts become larger but basically stay
within the range of 4 km for SatZ smaller than 40°. For even larger SatZ (larger than 40°), the
magnitude of shifts can reach 6 km for NOAA-17 scene and 8 km for MetOp-B scene. From
these results, it can be inferred that the SatZ has a considerable effect on both the magnitude
and uncertainty of the shifts in across-track direction. The larger SatZ generally contributes to
larger shifts and uncertainties in the across-track direction. Furthermore, it can be inferred that
the GAC data with SatZ less than 40° should be preferred in applications.

Compared to the shifts in the across-track direction (Figs. 8a-c), the shifts in the along-

track direction show smaller variability at each fixed SatZ (Figs. 8d-f). From Figs. 8d-e, it can
be seen that the shifts in the along-track direction are relatively stable at each level of SatZ for
SatZ smaller than 15°, but becomes more variable for greater SatZ. A similar phenomenon can
be observed in Fig. 8f, where the shifts are relatively stable with SatZ ranging from 20° to 35°,
but becomes more variable at each level of SatZ with its values larger than 35°. It is noteworthy
that the wide spread of shifts with SatZ less than 20° is mainly caused by cloud contamination.
These results confirm the influence of larger SatZ on the uncertainty of shifts in the along-track
directions. It is interesting to find that the magnitudes of NOAA-17 scene shifts with small SatZ
(less than 20°) are even larger than those with larger SatZ (larger than 20°) (Fig. 8d). On the
contrary, the magnitudes of MetOp-B scene shifts with smaller SatZ (20-35°) are smaller than
those with larger SatZ (larger than 35°) (Fig. 8f). Nevertheless, all three sensors have in
common that they do not show clear change with SatZ smaller than 20° for NOAA-17 and
smaller than 35° for MetOp-A and MetOp-B (Figs. 8d-f). For larger SatZ than these values,
shifts exhibit a slightly decreasing trend for NOAA-17 (Fig. 8d) and an increasing trend for
MetOp-B (Fig. 8f). From these results, it can be stated that the influences of large SatZ on the
magnitude of shifts in the along-track direction are probably intertwined with other factors.

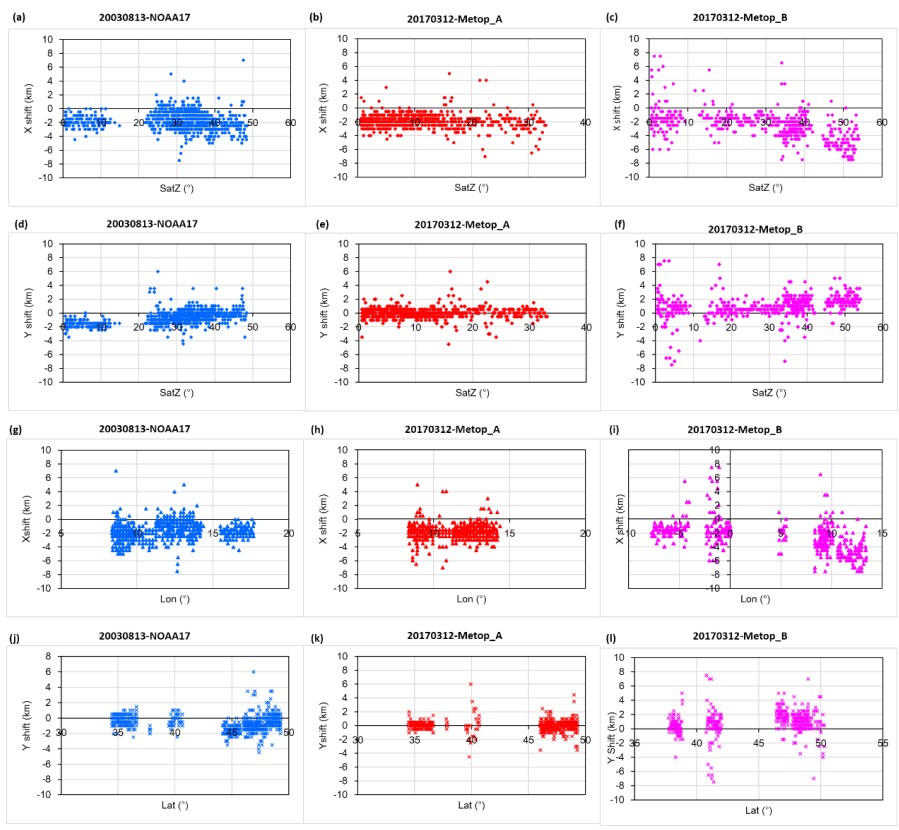


**Figure 8.** Influence of SatZ (a-c), longitude (d-f), and latitude (g-i) on the geolocation results of NOAA-17 (left), MetOp-A (middle) and MetOp-B (right) scenes.

**Table 5**. The mean shift for each range of SatZ in the across-track direction. The unit of the shift is km.

| SatZ | 0°-10° | 10°-20° | 20°-30° | 30°-40° | 40°-50° | 50°-60° |
|---|---|---|---|---|---|---|
| NOAA-17 | -1.84 | -1.84 | -1.32 | -1.66 | -2.27 | |
| MetOp-A | -1.87 | -1.80 | -2.06 | -2.62 | | |
| MetOp-B | -1.29 | -1.45 | -1.75 | -2.71 | -3.95 | -4.93 |

From Figs. 8g-i, it can be found that the variation of shifts (in the across-track direction) with longitude largely depends on the situation. For NOAA-17, the shifts tend to be smaller with the longitudinal range of 10°-15° and become larger outside this range (Fig. 8g). The MetOp-A scene does not show apparent change with longitude between 8° and 15° and neither does MetOp-B within the range of -8°-0°. However, MetOp-B presents a clear decreasing trend (an increasing trend in magnitude) for longitudes larger than 5°. Given the fact that the latitude of the nadir area is distributed between 10°-15° for NOAA-17, 8°-15° for MetOp-A, and -8°-0° for MetOp-B (Figs. 1b and d, Figs. 2b and d), it can be concluded that the influence of longitude on the shifts in the across-track direction is related to the longitude of nadir area of



the satellite, as it shows almost no influence in the nadir area. The influence increases with the
difference of the longitude relative to that of the nadir area. This is well understandable, as the
influence of longitude is equivalent to that of SatZ in the across-track direction.
The variation of the shifts (in the along-track direction) with latitude also depends on the
situation (Figs. 8j-l). The magnitudes of shifts with larger latitude (larger than 45°) are generally
greater than those with smaller latitude (less than 40°) on the NOAA-17 (Fig. 8j) and MetOp-
B scene (Fig. 8l). This is not visible for the MetOp-A scene (Fig. 8k), where the shifts exhibit
almost no change with latitude.  This can be attributed to the fact that the clock drift errors are
corrected more thoroughly for MetOp-A satellite than NOAA-17 and MetOp-B satellites.
Furthermore, the MetOp satellites have an on-board stabilization to keep them in the right
position and orientation in orbit compared to the NOAA satellites.

## 5  Conclusions

The geometric accuracy of satellite data is crucial for most applications as geometric
inaccuracy can bias the obtained results. Therefore, the assessment of the geolocation accuracy
is important to provide satellite data of high quality enabling successful applications. In this
study, a correlation-based patch matching method was proposed to characterize and quantify
the AVHRR GAC geo-location accuracy. This method presented here yields significant
advantages over existing approaches and enables achieving a subpixel geo-positioning accuracy
of coarse resolution scenes. It is free from the impact of false detection due to the influence of
mixed pixels, not limited to a certain landmark (e. g. shoreline) and therefore enables a more
comprehensive geometric assessment. This method was utilized to characterize the geolocation
accuracy of AVHRR GAC scenes from NOAA-17, MetOp-A, and MetOp-B satellites.
The study is based on several ROIs comprising numerous patches over different land cover
types, latitudes, and topographies. The scenes from these satellites all present West shifts in the
across-track direction, with an average shift of -1.69 km and a StdDev of 1.32 km for NOAA-
17, -1.9 km and 1.1 km respectively for MetOp-A, and -2.56 km and 2.19 km respectively for
MetOp-B. In regard to the shifts in the along-track direction, NOAA-17 generally shows South
shifts with an average of -0.7 km and a StdDev of 1.01 km. By contrast, the MetOp-B mainly
present North shifts with an average of 0.96 km and a StdDev of 1.70 km. The MetOp-A scene
shows a distinct advantage over NOAA-17 and MetOp-B in the along-track direction without
obvious shifts, indicated by the average of -0.02 km and a StdDev of 0.79 km. Generally, the
MetOp-B scene is inferior to NOAA-17 and MetOp-A scenes, with larger shifts and
uncertainties in both directions. Despite the variation of shifts due to various factors (e. g. SatZ,
topography), more than 90 percent of the AVHRR GAC data across-track errors are within $\pm$
3 km for NOAA-17 and MetOp-A, and $\pm 5.5$ km for MetOp-B. Along-track errors are within
$\pm 2$ km for NOAA-17, $\pm 1$ km for MetOp-A, and $\pm 3$ km for MetOp-B for more than 90
percent of the test data. It is important to note that since these satellites show different shifts,





using the combined data from NOAA-17 and MetOp will result in additional uncertainty in
time series applications.
From the results above, it can be found that the geolocation accuracy in the along-track
direction is always higher and with less uncertainties than the across-track direction, which is
consistent with previous related studies. This is understandable since the GAC dataset from the
ESA cloud CCI project has been corrected for clock drift errors, but has no ortho-correction,
which is not feasible due to the onboard sampling characteristics. SatZ plays a decisive role in
determining the magnitude as well as the uncertainty of the shifts in the across-track direction.
Larger SatZ generally induce greater shifts and uncertainties in this direction. The combined
effect of SatZ and topography on geolocation accuracy in the across-track direction has also
been shown. And significant terrain effects appear only in the case of large SatZ (>40° for this
study). It is important to note that the effect of SatZ on the magnitude and uncertainty of shifts
in the along-track direction is not negligible. But this effect is likely to be intertwined with other
factors. The impact of longitude on the shifts in the across-track direction is equivalent to that
of SatZ, while the effect of latitude is related to the degree of how the clock drift errors are
corrected. It was found that the clock drift errors are more thoroughly corrected for MetOp-A
than NOAA-17 and MetOp-B.
Although this assessment was only conducted for a single scene of each satellite, it
provides an important preliminary geolocation assessment for AVHRR GAC data. It is a first
step towards a more precise geolocation and thus improves application of coarse-resolution
satellite data. For instance, it identifies the threshold of SatZ under which the GAC data should
be preferred in applications. Furthermore, the CPMM geolocation assessment method proposed
by this study is also applicable to other coarse-resolution satellite data.

## Data availability

The AVHRR GAC test data in this paper draw on datasets from ESA CCI cloud project
(http://www.esa-cloud-cci.org/) where is also the data availability indicated (Stengel et al.,
2017). And the MOD13A1 V006 data can be downloaded via
https://ladsweb.modaps.eosdis.nasa.gov/ (Didan, 2015).

## Author contributions

Xiaodan Wu was responsible for the main research ideas and writing the manuscript.
Kathrin Naegeli contributed to the data collection. Stefan Wunderle contributed to the
manuscript organization. All the authors thoroughly reviewed and edited this paper.

## Competing interests

The authors declare that they have no conflict of interest.



## Acknowledgments

The authors are grateful to the ESA CCI (Climate Change Initiative) cloud project team (Dr. Martin Stengel, Dr. Rainer Hollmann) to make the data sets available for this study. This work was supported by the National Natural Science Foundation of China (41801226).

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
