# Peer review of "Geometric accuracy assessment of global coarse resolution satellite data sets: a study based on AVHRR GAC data at the subpixel level"

_Earth System Science Data, 2019_

## Referee Comment (RC1) · Anonymous Referee #1 · 5 Sep 2019

SUMMARY: This study assessed geometric accuracy of global coarse resolution satellite data sets via a Correlation-based Patch Matching Method (CPMM). This study aimed to quantify the AVHRR Global Area Coverage (GAC) at the subpixel level from three different satellite products from the NOAA-17 and the Meteorological Operational Satellites (MetOp-A and -B). This study selected multiple study regions to evaluate the potential influence factors such as satellite zenith angles, latitude, longitude, and elevation. The findings of this study supported that CPMM succeeded in quantifying uncertainties of in different satellite data and identifying key influence factors/sources in their uncertainties. However, there is a major comment about the robustness of this method for more other cases. In this study, this method was evaluated for the sin-

gle dates (August 13, 2003 for NOAA-17 and March 12, 2017 for MetOp-A and -B). Based on the results from a single date, the robustness of this method is still in question. As authors mentioned in the manuscript (line 34-36), an advantage of AVHRR sensors is that they have a long-term data since 1980s, which enables us to analyze it at the climate time scales. The findings of this study is more likely a case study of the geometric accuracy assessment for a single satellite imagery data. Another major comment is related to the scientific representation, particularly the figures. What do the Y-axes of Figure 5, 6, and 7 represent? It is not clear what 0.8 (in Figure 6 (a)) or 4 (in Figure 6 (e)) meant assuming that the sum of the density of all the bins should be either 1 or 100. Please clarify the maximum value of the density. Secondly, Figures 5, 6, and 7 showed the histograms along the shifts in the along-track and across-track directions ranging from -8 and +8 kilometers with an interval, 500 meters over different study regions. Can authors show the changes in correlations along the shifts in the along-track and across-track directions as well? Here is a suggestion that authors can plot bi-histograms of 1,089 (33 x 33) samples along the shifts (the x-axis; blue for the along-track direction and red for the across-track direction) and correlations (the y-axis). Based on Figure 3, the correlations are various depending on the shifts in the directions. It might be worth showing these changes along the shifts as well. Based on these major comments, the topic and scope of this manuscript are well fit to ESSD but it is publishable after major revision. Minor comments are provided below:

Minor comments:

Line 9: Global Area Coverage (GAC), not GAC (Global Area Coverage).

Line 34: "are" instead of "is"

Line 61: What does "reduced resolution" mean here? Maybe "coarse resolution"?

Line 81: What are "certain conditions"? Please explain it in more details.

Line 82-83: "... but *it* depends on ..." Also, is this sentence based on previous stud-

ies? If then, please cite the reference. Line 100-101: The sentence is not clear. Maybe "to test some satellite data from NOAA-17, ..."

Line 165-166: Please consider to change the ROI numbers. For example, for the mountainous areas, the ROI numbers are 1, 2, and 3 (currently, 2, 4, and 6, respectively). For the flat regions, the ROI numbers are 4, 5, and 6 (currently, 1, 3, and 5).

Figure 2: Please use different ROI labels since they are different from regions of interest in Figure 1. It is confusing if the numbers are used for ROI labels in Figure 1 and Figure 2. I suggest authors to use letters for ROI indicators (e.i., A, B, C, D, E, and F) in Figure 2.

Line 187: "CCM", not "CGM".

Line 187: land-sea fraction method (LFM) since the full name of LFM appeared in line 76.

Line 187-191: These sentences are redundant. Please remove them.

Figure 3: Please use white filled boxes or arrows, instead of blue filled boxes or arrows.

Line 218-219: I have a major concern about the robustness of this method for other regions and other seasons. Please see my first major concern above.

Figure 4: Please use a larger range of the color scale. It is hard to find the grid cell/location of the maximum correlation.

Line 275-277: Mean and standard deviation are parametric statistics of the data that are from a normal distribution. However, the shifts might be not well fitted to a normal distribution (based on Figure 3). I suggest authors to use the median of the shifts and their first and third quartiles.

Line 282-283: Please rewrite this sentence in the order of ROIs 5 and 6.

Line 284 and 286: "ROIs *1, 3, 4, and 7*" Is there any reason to keep the order of ROIs

(7, 3, 1, 4)?

Line 291: "ROIs 2, *5, and 6*"

Figure 5: Please state what the blue and red histograms represent.

Table 2, 3, and 4: Please add the elevations of ROIs. This information will be helpful for readers to understand the impact of elevation on the accuracy.

Figure 8: "SatZ (*a-f*), longitude (*g-i*), and latitude (*j-l*)"

Line 495-496: Please remove this sentence.

Line 499: "... within the range *between* -8° *and* 0° *(Fig. 8 h and i, respectively)*".

Line 558-559: As authors mentioned, this study was conducted only for a single scene. It questions: 1) is this study novel enough to contribute to various applications of the satellite data used in this study (particularly for climate research)? Or, was conducted a comprehensive assessment for the robustness of this method. The current results are more likely based on a case study for geometric accuracy assessment of coarse resolution satellite datasets.

---

## Referee Comment (RC2) · Anonymous Referee #2 · 17 Sep 2019

This paper describes the work undertaken by the authors to assess the geolocation accuracy at the subpixel level of AVHRR Global Area Coverage data from NOAA-17, MetOp-A and MetOp-B satellites. The paper is comprehensive and generally well written, with sufficient figures to follow the work that is described. The authors used a coregistration method based on reference NDVI data from MODIS. The authors use NDVI from NOAA-17, MetOp-A and MetOp-B satellites (using visible and near IR bands), and use the described Correlation-based Patch Matching Method to assess the sub-pixel geo-location accuracy in both the along-track and cross-track directions. Six regions of interest from Europe and Africa were selected for analysis, and included different land cover and terrain characteristics. The effect of large satellite zenith angles was also examined. Results are presented as mean cross-track and along-track shifts along with a standard deviation for each of the satellites applied to each of the regions of interest. The analysis was thorough, and I cannot suggest any further work needed for the paper.

I have a few specific comments for the paper: 1) Provide a reference or two on the land-sea fraction method mentioned on page 3. 2) When introducing figures 1 and 2, point out the color bar for SatZ and that the white line represents small SatZ along the satellite path. This will be helpful to the reader. 3) The figure 8 caption is not correct. The first two rows are SatZ cross-track and along-track (a-c) and (d-f). Longitude should be (g-i). Latitude should be (j-l).

––––––––––––––––––––––––––––––––

---

## Author Comment (AC1) · 1 Nov 2019

Thank you for spending your time on my manuscript and giving the opportunity to revise it. And we also thank you for your valuable comments as they actually improve the paper's quality. In this document, we describe how we address the reviewer's comments. The reviewer's comments are marked in black color, however, our reply is marked in blue color. SUMMARY: This study assessed geometric accuracy of global coarse resolution satellite data sets via a Correlation-based Patch Matching Method (CPMM). This study aimed to quantify the AVHRR Global Area Coverage (GAC) at the subpixel level from three different satellite products from the NOAA-17 and the

Meteorological Operational Satellites (MetOp-A and -B). This study selected multiple study regions to evaluate the potential influence factors such as satellite zenith angles, latitude, longitude, and elevation. The findings of this study supported that CPMM succeeded in quantifying uncertainties of in different satellite data and identifying key influence factors/sources in their uncertainties. However, there is a major comment about the robustness of this method for more other cases. In this study, this method was evaluated for the single dates (August 13, 2003 for NOAA-17 and March 12, 2017 for MetOp-A and -B). Based on the results from a single date, the robustness of this method is still in question. As authors mentioned in the manuscript (line 34-36), an advantage of AVHRR sensors is that they have a long-term data since 1980s, which enables us to analyze it at the climate time scales. The findings of this study is more likely a case study of the geometric accuracy assessment for a single satellite imagery data. Re: We admit that this study is a case study for geometric accuracy assessment of coarse resolution satellite datasets. In fact, a comprehensive geometric assessment of AVHRR GAC dataset over a long time series is not the focus of this paper. Instead, this study aims to propose a geometric assessment method specified for coarse resolution satellite datasets. We insist that this method is novel and robust to support a comprehensive geometric assessment. This can be explained from following aspects: 1. The traditional methods do not work well with coarse resolution satellite data due to the difficulty of detecting robust features. Even for stable lake or sea shorelines, the results suffer from the errors caused by false detection resulted from the effect of mixed pixels. Furthermore, such evaluations are often limited to certain landmarks and thus cannot support a comprehensive geometric assessment. Therefore, a more appropriate method specified for coarse resolution satellite dataset should be developed to enable a more accurate and comprehensive geometric assessment. Given the difficulty of detecting GCP on a coarse resolution image, this study has brought forward a way of thinking in another view, namely simulating the displacements with the reference map at a certain step length in different directions (within the range of ±8 km), and then checking whether the image and the reference map match the best.

[Figure]

The geometric accuracy of the image was indicated by the simulated displacement with the best correlation. The basic principle is that when the simulated displacements is equal to the geometric errors of the coarse-resolution image, the correlation between the image and the reference map is the largest because the spatial distribution characteristics of their spectrums are completely consistent. Considering the influence of satellite zenith angle, topography, landcovers, latitudes and longitudes on geometric accuracy, the image is divided into many patches. And the final geometric accuracy of the image was determined by statistically summarizing the measured displacements of a very large number of patches (from the image). This method has three advantages:  It works directly on pixel values, not a certain landmark. Therefore, it does not suffer from the errors caused by false detection due to the effect of mixed pixels and is not limited to certain landmarks. Consequently, it enables a more accurate and more comprehensive geometric assessment.  The method itself is based on correlation calculation between the image and the reference map. It is obvious that the method is not dependent on regions or seasons. As long as the reference image and the ROI satisfy certain requirements, the method can be applied to other regions (except for homogeneous surface like water and desert) and other seasons.  Since the method can be applied continuously in space, it provides the opportunity to study the effect of the influential factors (SatZ, topography, etc.) on geometric accuracy. 2. Although the current study is based on single dates, the selected ROIs cover different latitudes and longitudes, SatZ angles, land covers, as well as varying topographies, which represent typical influential factors on geometric accuracy. The surface conditions of these ROIs are very complicated, which is similar to most of the land surfaces on earth. Therefore, it is reasonable to believe that this method is robust and can be applied to other regions comprehensively. However, it is important to note that the method cannot work well over homogeneous surfaces such as water and desert. Because in such a situation, the correlations between the patches of the image and the reference map are always similar in any simulated displacement cases. In order to clarify this point, we have explained the possibility of applying this method to other regions and clarified that

the method cannot apply to homogeneous surfaces in Line 569-573 as "Although this assessment was only conducted for a single scene of each satellite, the highly variable ROIs take the influential factors of geometric accuracy well into account. Therefore, the presented conclusions are transferable to other regions or seasons. However, it is noteworthy that this method is not applicable to homogeneous surface (e.g., water, desert), where the correlations are almost the same in any simulated displacement cases.". As mentioned by the reviewer, the advantage of AVHRR sensors is the availability of data over multiple decades. Therefore, the AVHRR GAC data should be evaluated for all NOAA and MetOp platforms throughout the lifetime of the dataset on the global scale, which is particularly important for climate research. However, this is not the focus of this study. Despite the limited dataset used in this study, it still provides an important preliminary geolocation assessment of AVHRR GAC dataset by indicating its geometric accuracy in different situations (e.g., satellite platform, SatZ, topography). More importantly, it identifies the threshold of SatZ under which the GAC data should be preferred in applications. To clarify our main aim of the paper, and prevent deception of readers, we adjusted the manuscript at the following, prominent positions: -Title: we deleted the word "global". The new title reads as follows: "Geometric accuracy assessment of coarse resolution satellite data sets: a study based on AVHRR GAC data at the subpixel level". -Abstract: We clarified our main aim: "In this study, a Correlation-based Patch Matching Method (CPMM) was proposed to characterize and quantify the geo-location accuracy at the subpixel level for satellite data with coarse resolution, such as AVHRR GAC dataset. This method is neither limited to landmarks nor suffers from errors caused by false detection due to the effect of mixed pixels caused by a coarse spatial resolution, and thus enables a more robust and comprehensive geometric assessment than existing approaches." This method is expected to yield significant advantages over existing approaches in characterizing the geometric accuracy of coarse resolution dataset and enable a more comprehensive and robust geometric assessment at the sub-pixel level. We sincerely think this paper has the potential and practical values to be introduced to readers. We believe that as time goes

on, the long-term AVHRR GAC data can be more comprehensively evaluated at the climate time scales.

Another major comment is related to the scientific representation, particularly the figures. What do the Y-axes of Figure 5, 6, and 7 represent? It is not clear what 0.8 (in Figure 6 (a)) or 4 (in Figure 6 (e)) meant assuming that the sum of the density of all the bins should be either 1 or 100. Please clarify the maximum value of the density. Re: As noticed by the reviewer, it is important to understand exactly what the vertical scale is representing and how that is affected by the choice of bins when reading a histogram. If the bins are of equal size, a rectangle is erected over the bin with height proportional to the frequency—the number of cases in each bin. A histogram may also be normalized to display "relative" frequencies. In such a case, the sum of the height of all the bins should be equal to 1. However, in the Figures 5, 6, and 7, we can see that the histograms are with non-uniform (unequal) class widths. In such a case, the area of the bin is proportional to the class frequencies, and the ordinate is labelled density instead of frequency.

Namely, it is the area of each bin that denotes the relative frequency of each class, not the height. And the total area of the histogram is equal to 1. Since the height of the bins indicate the density (not frequency), the sum of the height of all bins is unequal to 1 or 100. And the maximum value of the density is not fixed, depending on the widths of the bins. More detailed information about histograms with non-uniform width can be found at https://www.datamentor.io/r-programming/histogram/ and https://wtmaths.com/histograms_unequal_intervals.html. In order to make the histograms more understandable to readers, we have explained the meaning of the "density" in the caption of the figure as "For histograms, the heights of the bars indicate the density. In this case, the area of each bar is the relative frequency, and the total area of the histogram is equal to 1." in Line 310-311.

Secondly, Figures 5, 6, and 7 showed the histograms along the shifts in the along-track and across-track directions ranging from -8 and +8 kilometers with an interval, 500 meters over different study regions. Can authors show the changes in correlations along the shifts in the along-track and across-track directions as well? Here is a suggestion that authors can plot bi-histograms of 1,089 (33 x 33) samples along the shifts (the x-axis; blue for the along-track direction and red for the across-track direction) and correlations (the y-axis). Based on Figure 3, the correlations are various depending on the shifts in the directions. It might be worth showing these changes along the shifts as well. Based on these major comments, the topic and scope of this manuscript are well fit to ESSD but it is publishable after major revision. Re: We would like to thank the reviewer here for giving such a suggestion. However, we would like to point out that the histograms in Figures 5, 6, and 7 are not derived from the 1,089 (33 x 33) simulated displacement samples. In fact, they indicate the distribution of the measured displacements of all patches within each ROI, which is used as the indicator of geometric accuracy of AVHRR GAC dataset. As explained before, the basic idea of the method is simulating the displacements with the reference map at a step of 500 m in different directions (within the range of ±8 km), and then checking where the patch on AVHRR image and the patch on reference map match the best among the $33\times33$ simulated cases. The simulated displacement with the largest correlation indicates the geometric accuracy (the measured displacement) of this patch. Given that the geometric accuracy varies with SatZ, topography, etc., the image is divided into many small patches. For each patch, there are $33\times33$ simulated displacement samples considering different combinations of X- and Y- simulated displacements (as shown in Figure 4). The final geometric accuracy of one image was determined by statistically summarizing the measured displacements of a very large number of patches from the image (as shown in Figures 5, 6, and 7). And this is the goal of the method. As regard to the correlations of the $33\times33$ simulated samples, it is their relative magnitude that is of interest, not the absolute values, as the goal is to identify the sample with the largest value among these $33\times33$ simulated samples. Since the reviewer is interested in the changes in correlations along the simulated shifts in the along-track and across-track directions, we presented the variations of correlations in the form of image as follows, because

the images are more intuitive than bi-histogram to show the magnitude of correlation of different simulated displacements (in the four quadrants: -X,-Y, +X, +Y). Since there are so many patches for each scene (759, 632, and 469 for NOAA17, MetOp-A, and MetOp-B, respectively), it is impractical to include the results of all patches here. Thus, only part of the patches of ROI 6 on NOAA 17 scene are shown for conciseness.

Figure. The variations in correlations of the 33×33 simulated displacements. Each subfigure represent the result of one patch. The center of the image indicates the case of no shift in two directions. Each grid represents a simulated shift combination case, which is indicated by the location of the grid away from the center in the along-track and across-track direction. The geolocation errors can be transferred into distances in kilometer (km) by multiplying the location of a grid with 500 m. As noticed by the reviewer, the correlations are varying depending on the simulated displacements in the two directions. More importantly, the variation of correlations along the simulated shifts in the along-track and across-track directions shows very large differences from one patch to another. There is not a fixed relationship between the correlations and the simulated shifts in both directions. Actually, this is reasonable. Because when the simulated displacement is not equal to the actual geometric errors of the patch, their underlying surface is not the same. As a result, the spatial distribution spectral characteristics are not consistent between the patch on the image and that on the reference map. And the degree of their agreement (namely, the correlation) along with simulated shifts in two directions is random and differs among different patches. This can also be confirmed by the above figure. In fact, the changes in correlations are related to the proximity of the simulated displacement to the actual displacement of the patch (taken from AVHRR GAC scene). From the above figure, it can be seen that the correlation appears a maximum at a certain location, and then becomes gradually smaller with increasing distance from that location. The location with the maximum correlation indicates the actual displacement of this patch. The farther away the simulated displacement are from that location, the smaller the correlations. In conclusion, the change in correlations do not show a clear trend along the simulated shifts in the along-track

and across-track directions. Instead, the correlations show a decreasing trend with the distance deviated from the location corresponding to the actual geometric error of the patch (AVHRR GAC). In order to clarify this point, we have added one sentence as "It can be seen that the correlation appears a maximum at a certain location, and then becomes gradually smaller with increasing distance from that location. The location with the maximum correlation indicates the actual displacement of this patch." in Line 261-263 in the new manuscript.

Minor comments are provided below: Line 9: Global Area Coverage (GAC), not GAC (Global Area Coverage). Re: "GAC (Global Area Coverage)" has been changed to "Global Area Coverage (GAC)". Line 34: "are" instead of "is" Re: This error has been corrected. Line 61: What does "reduced resolution" mean here? Maybe "coarse resolution"? Re: Local Area Coverage (LAC) data has a spatial resolution of 1.1 km. Global Area Coverage (GAC) data has a spatial resolution of 4 km. Here, "reduced resolution" means lower spatial resolution. In order to make it clearer to readers, we have changed it to "coarser resolution" in the new manuscript. Line 81: What are "certain conditions"? Please explain it in more details. Re: We agree, the term "certain conditions" is not clearly defined and the explanation followed in the subsequent sentences wasn't clear enough linked. We thus re-phrased this section in Line 82-90 as follows: "The abilities of these three methods in characterizing the geometric errors are limited and dependent on different, method-dependent factors. Whereas, the CCM is subject to the structure of coastline, and the LFM depends on the accuracy of the land-sea model but shows advantages on complex coastlines (Han et al., 2016). The coregistration method is usually applied to high-resolution visible and infrared images (Wang et al., 2013; Wolfe et al., 2013) as it relies on individual objects/landmarks in both datasets. However, when it comes to coarse resolution data with several kilometers' pixel size, the main difficulties arise from false detection due to the effect of mixed pixels, which hampers the application of the existing methods. An approach assessing the geolocation accuracy of coarse resolution satellite data is thus strongly needed. " Line 82-83: "... but *it* depends on ..." Also, is this sentence based on previous
studies? If then, please cite the reference. Re: This sentence has been rephrased as "......, and the LFM depends on the accuracy of the land-sea model but shows advantages on complex coastlines (Han et al., 2016)" in Line 84-85. This sentence is based on a previous study, which has been cited in the new manuscript. Line 100-101: The sentence is not clear. Maybe"to test some satellite data from NOAA-17, ..." Re: This sentence has been rephrased as "This method is tested using satellite data from NOAA-17, MetOp-A, and MetOp-B, respectively." in Line 105-106. Line 165-166: Please consider to change the ROI numbers. For example, for the mountainous areas, the ROI numbers are 1, 2, and 3 (currently, 2, 4, and 6, respectively). For the flat regions, the ROI numbers are 4, 5, and 6 (currently, 1, 3, and 5). Re: We would like to thank the reviewer here for giving such a suggestion. As pointed out by the manuscript, the ROIs are distributed over different latitudes and longitudes, different geographic locations, different SatZ angles, different land covers, as well as different topographies in order to explore the potential impact factors related to geolocation accuracy. The topography is only one of factors that need to be considered. Therefore, the ROIs are not numbered in the order in which only mountainous/flat regions are considered. In fact, the number of the ROI is just used as a label, and we would like to keep the ROI numbers in Figure 1 in the new manuscript. Figure 2: Please use different ROI labels since they are different from regions of interest in Figure 1. It is confusing if the numbers are used for ROI labels in Figure 1 and Figure 2. I suggest authors to use letters for ROI indicators (e.i., A, B, C, D, E, and F) in Figure 2. Re: As suggested by the reviewer, we have used letters (i.e., a, b, c, d, e, and f) for ROI indicators in Figure 2. Line 187: "CCM", not "CGM". Re: "CGM" has been revised as "CCM" Line 187: land-sea fraction method (LFM) since the full name of LFM appeared in line 76. Re: This has been corrected. Line 187-191: These sentences are redundant. Please remove them. Re: This sentence has been removed in the new submission. Figure 3: Please use white filled boxes or arrows, instead of blue filled boxes or arrows. Re: We have used white filled boxes or arrows in the figure. Line 218-219: I have a major concern about the robustness of this method for other regions and other seasons. Please see

my first major concern above. Re: As explained above, the method itself is based on correlation calculation between the patches taken from AVHRR GAC image and the patches with simulated displacement taken from the reference map. The geometric error of one patch was indicated by the simulated displacement with the largest correlation. And the final geometric accuracy was determined by statistically summarizing the measured displacements of a very large number of patches (759, 632, and 469 for NOAA17, MetOp-A, and MetOp-B, respectively) from the image. There are four basic requirements of the method: First, the reference image should have much higher spatial resolution than the image that is to be evaluated (e.g., 500m vs. 4 km in this study). Second, referenced image itself should have high geometric accuracy. Third, with consideration of temporal variations of the spectral properties of the Earth surface, the date of the reference image should be equal or close to the image to be evaluated. Last but not the least, the individual patches should present a spatially heterogeneous distribution of surface characteristic and thus varying spectra. Homogeneous surface (e.g., water, desert) should be avoided, as for these surfaces, the correlations are always large and similar in any simulated displacement cases. Consequently, it is hard to identify the actual displacement of the patch on the AVHRR GAC image. It is obvious that the method works directly on pixel values. It is thus neither dependent on regions nor on seasons. As long as the reference image and the ROI satisfy the above requirements, the method can be applied to other regions and seasons. In order to clarify this point, we added one sentence as "Although this assessment was only conducted for a single scene of each satellite, the highly variable ROIs take the influential factors of geometric accuracy well into account. Therefore, the presented conclusions are transferable to other regions or seasons. However, it is noteworthy that this method is not applicable to homogeneous surface (e.g., water, desert), where the correlations are almost the same in any simulated displacement cases." in Line 569-573. Figure 4: Please use a larger range of the color scale. It is hard to find the grid cell/location of the maximum correlation. Re: The color scale has been optimized by algorithmic procedures, and only the minimum and maximum were covered. The difficulty to find

the maximum correlation in Figure 4 lies in the fact that the correlations of simulated displacements which are close to the actual displacement of one patch are similar and approach the maximum correlation. But this will not affect the final results because the maximum correlation is identified through the algorithm, not from the image in the figure. The Figure 4 just shows an example of one patch for illustration purposes. The main aim of Figure 4 is to help readers better understand the process of determining the displacement of one patch. Line 275-277: Mean and standard deviation are parametric statistics of the data that are from a normal distribution. However, the shifts might be not well fitted to a normal distribution (based on Figure 3). I suggest authors to use the median of the shifts and their first and third quartiles. Re: We would like to thank the reviewer here for your valuable advice. According to Han et al. (2016), Sultan et al. (2015), Aguilar et al. (2013), and Bicheron et al. (2011), geometric accuracies are generally measured as the mean shifts and standard deviations, with the former representing the magnitude of the displacements and the latter indicating the uncertainty. The small standard deviation values indicate a high reliability for the geometric accuracy assessment. In line with previous studies, the mean and standard deviation were adopted in this study. Since the reviewer is interested in the result with the median of the shifts and their first and third quartiles, the boxplots of the displacements in the across-track and along-track directions are presented here.

Figure. The boxplots of shifts in the across-track (X) and along-track (Y) directions over different regions for NOAA-17, MetOp-A, and MetOp-B scenes. From the above figure, it can be seen that the scenes from these satellites all present West shifts in the across-track direction. In regard to the shifts in the along-track direction, NOAA-17 generally shows South shifts. By contrast, the MetOp-B mainly present North shifts. And the MetOp-A scene shows a distinct advantage over NOAA-17 and MetOp-B in the along-track direction without obvious shifts. The geolocation accuracy in the along-track direction is always higher and with less uncertainties than the across-track direction given their closeness to 0 and more centralized distribution. In fact, the conclusions based on the boxplots are consistent with those based on the histograms with mean

and standard deviations shown in the manuscript. Moreover, the histograms are more straightforward to show the distribution of displacements at different levels. For these reasons, we would like to keep the mean and standard deviation (as well as the histograms) as the indicators of geometric accuracy in the new manuscript. Reference: Aguilar, M. A., Salda?a, María del Mar, & Aguilar, F. J. . (2013). Assessing geometric accuracy of the orthorectification process from geoeye-1 and worldview-2 panchromatic images. International Journal of Applied Earth Observation and Geoinformation, 21, 427-435. Bicheron, P. , Amberg, V. , Bourg, L. , Petit, D. , & Arino, O. . (2011). Geolocation assessment of meris globcover orthorectified products. IEEE Transactions on Geoscience and Remote Sensing, 49(8), 2972-2982. Han, Y. , Weng, F. , Zou, X. , Yang, H. , & Scott, D. . (2016). Characterization of geolocation accuracy of suomi npp advanced technology microwave sounder measurements. Journal of Geophysical Research: Atmospheres, 121(9), 4933-4950. Sultan, Aksakal , et al. "Geometric Quality Analysis of AVHRR Orthoimages." Remote Sensing 7.3(2015):3293-3319. Line 282-283: Please rewrite this sentence in the order of ROIs 5 and 6. Re: Here, the ROIs are sorted in ascending order according to the magnitude of displacements. Since the ROI6 has a smaller shifts of -1.33 than that of ROI5 (-1.35), the ROI6 rank in the front of ROI5. Line 284 and 286: "ROIs *1, 3, 4, and 7*" Is there any reason to keep the order of ROIs (7, 3, 1, 4)? Re: As explained above, the order of these ROIs agrees with the order of the magnitude of their displacements. Line 291: "ROIs 2, *5, and 6*" Re: This order agrees with the order of the magnitude of their shifts. Figure 5: Please state what the blue and red histograms represent. Re: We would like to thank the reviewer for this valuable advice. In order to make the figure more easy to read, we have explained the blue and red histograms in the caption of the figure as "Figure 5. The distribution of shifts in the across-track (X, represented by red histogram) and along-track (Y, denoted as blue histogram) directions over different regions for NOAA-17 scene. The unit of the shift is km.". Table 2, 3, and 4: Please add the elevations of ROIs. This information will be helpful for readers to understand the impact of elevation on the accuracy. Re: As suggested by the reviewer, the elevation information of these ROIs was extracted from

SRTM Digital Elevation Data Version 4 as following. It can be seen that the elevations are widely distributed for each ROI. Therefore, the averaged elevations for each ROI were calculated and added in Table 2, 3 and 4.

Figure. The histograms of elevations within these ROIs for the scenes from NOAA-17 and MetOp-A.

Figure. The histograms of elevations within these ROIs for the scenes from MetOp-B. Figure 8: "SatZ (*a-f*), longitude (*g-i*), and latitude (*j-l*)" Re: We have corrected this mistake in the new manuscript. The caption of Figure 8 has been revised as "Figure 8. Influence of SatZ on the geolocation accuracy in the across-track (a-c) and along-track (d-f) directions. (g-i) and (j-l) describe the influence of longitude and latitude on the geolocation accuracy in the across-track and along-track directions, respectively. The left column indicates results of NOAA-17 (blue), middle for MetOp-A (red), and right for MetOp-B (pink) scenes.". Line 495-496: Please remove this sentence. Re: This sentence has been removed as suggested by the reviewer. Line 499: "... within the range *between* -8 ◦ *and* 0 ◦ *(Fig. 8 h and i, respectively)*". Re: This sentence has been rephrased as ". . . . . . and neither does MetOp-B within the range between -8° and 0° (Fig. 8 h and i, respectively)" in Line 509-510 in the new manuscript as suggested by the reviewer. Line 558-559: As authors mentioned, this study was conducted only for a single scene. It questions: 1) is this study novel enough to contribute to various applications of the satellite data used in this study (particularly for climate research)? Or, was conducted a comprehensive assessment for the robustness of this method. The current results are more likely based on a case study for geometric accuracy assessment of coarse resolution satellite datasets. Re: We admit that the current results are based on a case study for geometric accuracy assessment of coarse resolution satellite datasets. We now clearly state this in the revised Title and Abstract, please see our answer above. As mentioned earlier, a comprehensive geometric assessment of AVHRR GAC dataset over a long time series was not the focus of this study. Instead, this study aims to propose a geometric

assessment method specified for coarse resolution satellite datasets. However, we insist on that the method is novel to support a comprehensive geometric assessment for various applications and that the assessment of this method is comprehensive in this study. This can be explained from following aspects: 1. The traditional methods do not work well with coarse resolution dataset due to the fact that the detection of any robust feature fails. Even for the relatively stable lake and sea shorelines, errors may be introduced due to the false detection resulted from the effect of mixed pixels. Furthermore, such evaluations are often limited to certain landmarks, which cannot represent the overall performance of one scene. Because the geometric accuracy varies with satellite zenith angle, latitudes and longitudes, and topography, etc. As a result, the geometric accuracy of AVHRR GAC dataset has not yet been adequately addressed, particularly over terrain areas. Since the commonly used methods are difficult to apply on coarse resolution imagery, this study has brought forward a way of thinking in another view, namely simulating the displacements with the reference map at a certain step length in different directions, and then checking where the image and the reference map match the best, the Correlation-based Patch Matching Method (CPMM). The basic principle is that when the simulated displacements is equal to the geometric errors of the coarse-resolution image, the correlation between image and reference map is the largest given that the spatial distribution characteristics of their spectrums are completely consistent. Since this method works directly on pixel values, it does not suffer from errors caused by false detection due to the effect of mixed pixels and is not limited to a certain landmark. Therefore, it enables a more accurate and more comprehensive geometric assessment. Moreover, it provides the opportunity to explore the influential factors on geometric accuracy. This method is expected to yield significant advantages over existing approaches and enables achieving a subpixel geo-positioning accuracy of coarse resolution datasets. 2. Despite that only single scene was used in this study, the ROIs are of great variety covering different latitudes and longitudes, SatZ angles, land covers, as well as varying topographies, which represent the several typical influential factors on geometric accuracy. Furthermore,

the surface condition of these ROIs is very complicated, which is similar to most of the land surfaces on earth (apart from homogeneous regions such as water and desert). Moreover, it is important to remember that the method is not dependent on regions or seasons as long as the reference image and the ROI satisfy the above-mentioned requirements. Therefore, it is reasonable to believe that the assessment of this method is comprehensive and robust. 3. As for the contributions to various applications of the AVHRR GAC data (particularly for climate research), we admit that the limited dataset does not provide a comprehensive geometric assessment of the AVHRR GAC dataset over the entire time span that the dataset is available. However, the method itself supports the geometric assessment in the long time series because it is independent of regions and seasons and could thus be applied to other NOAA sensors providing AVHRR GAC data. Given that there are so many factors influencing geometric accuracy (e.g., satellite platform, orbital drift, SatZ, terrain, latitude), the AVHRR GAC data should be evaluated for all NOAA and MetOp platforms throughout the lifetime of the dataset globally, which is particularly important for climate research. Therefore, a comprehensive time series analysis of the geometric accuracy of AVHRR GAC data is needed in the future. This study, however, still provides an important preliminary geolocation assessment by indicating the displacement of AVHRR GAC data in different situations (e.g., satellite platform, SatZ, topography).

Please also note the supplement to this comment:
https://www.earth-syst-sci-data-discuss.net/essd-2019-87/essd-2019-87-AC1-supplement.pdf
* * *

---

## Author Comment (AC2) · 1 Nov 2019

Thank you for spending your time on my manuscript and giving the opportunity to revise it. And we also thank you for your valuable comments as they actually improve the paper's quality. In this document, we describe how we address the reviewer's comments. The reviewer's comments are marked in black color, however, our reply is marked in blue color. This paper describes the work undertaken by the authors to assess the geolocation accuracy at the subpixel level of AVHRR Global Area Coverage data from NOAA-17, MetOp-A and MetOp-B satellites. The paper is comprehensive and generally well written, with sufficient figures to follow the work that is described.

[Figure]

The authors used a coregistration method based on reference NDVI data from MODIS. The authors use NDVI from NOAA-17, MetOp-A and MetOp-B satellites (using visible and near IR bands), and use the described Correlation-based Patch Matching Method to assess the sub-pixel geo-location accuracy in both the along-track and cross-track directions. Six regions of interest from Europe and Africa were selected for analysis, and included different land cover and terrain characteristics. The effect of large satellite zenith angles was also examined. Results are presented as mean cross-track and along-track shifts along with a standard deviation for each of the satellites applied to each of the regions of interest. The analysis was thorough, and I cannot suggest any further work needed for the paper. I have a few specific comments for the paper: 1) Provide a reference or two on the land-sea fraction method mentioned on page 3. Re: A reference (i.e. Bennartz, 1999) has been provided for land-sea fraction method in Line 85.

2) When introducing figures 1 and 2, point out the color bar for SatZ and that the white line represents small SatZ along the satellite path. This will be helpful to the reader. Re: We have pointed out the meaning of color bar and the white line in the captions of figures 1 and 2 in the new manuscript. The caption of figure 1 was revised as "Figure 1……….. (b) and (d) are their corresponding SatZ respectively, which is indicated by the color bar, with the white line representing small SatZ along the satellite path." in Line 182-183. And the caption of figure 2 was revised as "Figure 2……..(b) and (d) are their corresponding SatZ (indicated by the color bar), respectively. The white line in (d) represents small SatZ along the satellite path." in Line 186-187.

3) The figure 8 caption is not cor- rect. The first two rows are SatZ cross-track and along-track (a-c) and (d-f). Longitude should be (g-i). Latitude should be (j-l). Re: We have corrected this mistake in the new manuscript. The caption of figure 8 has been revised as "Figure 8. Influence of SatZ on the geolocation accuracy in the across-track (a-c) and along-track (d-f) directions. (g-i) and (j-l) describe the influence of longitude and latitude on the geolocation accuracy in the across-track and along-track directions,

respectively. The left column indicates results of NOAA-17 (blue), middle for MetOp-A (red), and right for MetOp-B (pink) scenes.".

Please also note the supplement to this comment:
https://www.earth-syst-sci-data-discuss.net/essd-2019-87/essd-2019-87-AC2-supplement.pdf

---

## Author Response (AR2)

**Author's response**

**Topical Editor Decision: Publish as is** (03 Feb 2020) by Prasad Gogineni

Comments to the Author:

The main concern of one of the reviewers is that the method presented is a case study of the geometric accuracy assessment of a single satellite image data. The authors have included the following statement their work and limitation.

"Although this assessment was only conducted for a single scene of each satellite, the highly variable ROIs take the influential factors of geometric accuracy well into account. Therefore, the presented conclusions are transferable to other regions or seasons. However, it is noteworthy that this method is not applicable to homogeneous surface (e.g., water, desert), where the correlations are almost the same in any simulated displacement cases."

They addressed reviewers' comments and suggestions, and improved the paper.

I recommend the publication of the paper.

Re: We would like to thank the editors for all you have done on the manuscript. And we also thank the reviewers here for providing so many valuable suggestions as they are very helpful for us to improve this paper's quality.

We have made a very thorough check of the final accepted manuscript and revised the spelling error in **Line 511**: "*latitude*" has been revised as "*longitude*".